# Self-Shielding of X-ray Emission from Ultrafast Laser Processing Due to Geometrical Changes of the Interaction Zone

**DOI:** 10.3390/ma17051109

**Published:** 2024-02-28

**Authors:** Julian Holland, Christian Hagenlocher, Rudolf Weber, Thomas Graf

**Affiliations:** Institut für Strahlwerkzeuge (IFSW), University of Stuttgart, Pfaffenwaldring 43, 70569 Stuttgart, Germany

**Keywords:** X-ray emission, ultrafast laser processing, self-shielding, laser plasma, X-ray safety, percussion drilling, raytracing, model calculation

## Abstract

Materials processing with ultrashort laser pulses is one of the most important approaches when it comes to machining with very high accuracy. High pulse repetition rates and high average laser power can be used to attain high productivity. By tightly focusing the laser beam, the irradiances on the workpiece can exceed 10^13^ W/cm^2^, and thus cause usually unwanted X-ray emission. Pulsed laser processing of micro holes exhibits two typical features: a gradual increase in the irradiated surface within the hole and, with this, a decrease in the local irradiance. This and the shielding by the surrounding material diminishes the amount of ionizing radiation emitted from the process; therefore, both effects lead to a reduction in the potential X-ray exposure of an operator or any nearby person. The present study was performed to quantify this self-shielding of the X-ray emission from laser-drilled micro holes. Percussion drilling in standard air atmosphere was investigated using a laser with a wavelength of 800 nm a pulse duration of 1 ps, a repetition rate of 1 kHz, and with irradiances of up to 1.1·10^14^ W/cm. The X-ray emission was measured by means of a spectrometer. In addition to the experimental results, we present a model to predict the expected X-ray emission at different angles to the surface. These calculations are based on raytracing simulations to obtain the local irradiance, from which the local X-ray emission inside the holes can be calculated. It was found that the X-ray exposure measured in the surroundings strongly depends on the geometry of the hole and the measuring direction, as predicted by the theoretical model.

## 1. Introduction

Laser material processing with pulse durations in the order of 1 ps is one of the most important approaches when it comes to machining with very high accuracy. The recent increase in the available average power by orders of magnitude allows to significantly increase the productivity of the process. At the same time, it is possible to achieve a very high precision in the order of µm because of the low thermal load of the surrounding material [1,2,3]. The optimum irradiance for most industrial applications lies in the range of 10^11^–10^13^ W/cm^2^, depending on the material, the process itself, and the pulse duration [4]. However, irradiances of up to 10^15^ W/cm^2^ can easily be achieved on the workpiece with the presently available laser systems using tight focusing.

It was shown that these high laser irradiances result in high-energy plasmas with high electron temperatures [3,5,6]. These electron temperatures are commonly given in electron Volts (eV) and are typically in the order of a few keV for laser plasmas, as treated in this paper [5]. The spectrum of the radiation emitted from these plasmas contains a significant fraction of photons with energies above 5 keV [7]. Under industrial conditions, i.e., room temperature and one standard atmosphere of ambient pressure, the transmission for photons with 5 keV energy through 35 cm of air is about 19% [8]. Therefore, this emission can reach the operator’s position.

The interaction of incident ultrashort laser pulses with the ablated material was well described in previous reports [3,5,6]. Most of the experimental studies on the emission of ionizing radiation from laser-produced plasmas were already carried out in the 1980s under high vacuum conditions [9,10,11,12,13,14], resulting in only a few publications considering industrial conditions [7,15,16,17,18]. With the increase in available power of recent ultrafast lasers, the topic is gaining interest in the area of industrial laser material processing, which has lead to the latest investigations about the influence of high pulse repetition rates [19], different materials [20] and the pulse duration [21] on the X-ray emission. There are several important differences between the two conditions, one being the air breakthrough (ionization of the gas atmosphere) at high intensities [22] when working in standard atmosphere, leading to defocusing of the incident laser beam and thereby reducing the local irradiance on the workpiece. Additionally, the air atmosphere usually present under industrial conditions is not transparent for X-ray photons with less than 2 keV of energy, leading to a shielding effect with the soft X-ray emission.

Most of the publications on the X-ray emission from laser processing with ultrashort pulses present dose rates that are measured in worst-case scenarios, where the laser beam is focused on the plane surface of a workpiece during the whole measurement time and applying the highest possible irradiance [7,15,23,24,25]. The corresponding results therefore do not represent the X-ray emission that is associated with common industrial processes. 

Processing with ultrafast lasers typically involves two effects which contribute to reducing the X-ray emission. First, a continuous increase in the area of the processed surface (e.g., in an evolving percussion-drilled hole) leads to a decrease in the local irradiance, which in turn leads to a reduced X-ray emission. Second, the material surrounding an evolving geometrical structure (e.g., drilled holes) shields the emitted X-ray radiation. The investigations presented in the following were performed to quantify these two effects for the case of percussion drilling. A model was developed to calculate the flux of the X-ray photons at the position of a detector, considering the changing local irradiance in the drilled hole during the process. The spatial distribution of the irradiance was computed by means of raytracing calculations assuming conically shaped holes. The theoretical predictions are shown to be in good agreement with experimental data.

## 2. Calculation of X-ray Emission

### 2.1. Raytracing in Laser Drilled Holes

Percussion-drilled holes can be described well through conically shaped geometry [4]. The main parameters are their depth $z_{h}$ and the diameter $d_{h}$ of the opening at the surface of the sample, as shown in Figure 1. The depth of the drilled hole increases with increasing number of applied laser pulses, while the diameter of the opening remains constant. The distribution of the irradiance J(z) on the walls of the evolving hole changes during the drilling process due to the changing geometry of the hole [26,27] and was calculated by means of raytracing for the present study.

The simulations were performed using a self-developed raytracing algorithm [26] considering a total of 500,000 rays and assuming laser parameters matching the ones which were used during the experiments. The absorption and the reflection of the individual rays are calculated using Fresnel’s equations, as well as the refraction index *n* and the extinction coefficient *k* of the processed material. Since stainless steel (1.4301) was processed using a Ti:Sa laser with a wavelength of 800 nm, pure iron with *n* = 2.87 and *k* = 3.36 [28] was assumed for the raytracing calculation. The distribution of the pulsed peak irradiance *J*(*z*) resulting from multiple reflections of the rays inside the hole is qualitatively sketched on the left in Figure 1.

### 2.2. Spectrum of the X-ray Emission

During the interaction of a laser pulse with the material, free electrons are created by the leading edge of the pulse [5]. The free electrons are further heated by the following part of the pulse due to different effects, such as inverse Bremsstrahlung and resonance absorption [11,12], resulting in a fraction of electrons with a high kinetic energy, which are commonly referred to as the hot electrons [5]. The high temperature of the hot electrons is responsible for the X-ray emission from the laser-induced plasma considered in this section.

Following the argumentation in [5,18], the spectral power of the X-ray radiation emitted from a given plasma is given by
(1)$$\frac{dP_{B}}{d\omega }=c_{e}\cdot e^{-\frac{\hbar \omega }{{k_{b}T}_{h}}},$$
where *k_b_* and ℏ are the Boltzmann and the reduced Plank constant, respectively, and
(2)$$c_{e}=V_{P}\cdot \frac{32\pi }{3}\cdot \sqrt{\frac{2\pi }{3m_{e}k_{b}T_{h}}}\cdot \frac{Z_{i}e^{6}n_{h}^{2}}{m_{e}c^{3}},$$
is a scaling factor depending on the volume *V_P_* of the emitting plasma, the degree of ionization *Z_i_*, the number density *n_h_* of the hot electrons in the plasma and their temperature *T_h_*. Further, *m_e_*, *e* and *c* are the mass and charge of an electron and the speed of light, respectively. All parameters were calculated following the argumentation in [18], except for the plasma volume *V_P_*, where the ablation area $\left(\pi \cdot r_{a}^{2}\right)$ was replaced by a small area *dA* on the wall inside the hole. The spectral distribution of the power is solely determined by *T_h_*, which, again, according to [18] scales with
(3)$$T_{h}\left(z\right)=0.476K\cdot {\left(\frac{\lambda^{2}}{µm^{2}}\cdot \frac{J(z)}{W/{cm}^{2}}\right)}^{-0.53},$$
for our experimental condition and where *λ* is the wavelength of the incident laser radiation and *J*(*z*) is the local irradiance. The combination of Equations (1) and (3) clearly reveals the dependency of the X-ray emission on the value of the local irradiance. Due to the multiple reflections of the incident radiation inside the drilled hole, the local irradiance further depends on the geometry of the hole, which is why the X-ray emission changes during the process due to the different effects, which are described in the following using the geometrical parameters shown in Figure 2.

The X-ray emission is influenced by two major effects related to the changes of the processed geometry. First, the local irradiance, and thus the spectral power of the X-ray emission, changes depending on the shape and the depth *z_h_* of the hole. The second effect is the shielding of the X-ray emission by the material surrounding the drilled hole. On the way from the emission inside the hole to a given point (detector/observer) outside of the hole, the radiation traverses the distance *s_shield_* through the material, as sketched by the purple line in Figure 2. The length *s_shield_* depends on the direction of observation, as characterized by the angle β and the depth *z_val_* of the considered X-ray emission. The transmitted power spectrum of low-energy X-ray photons through a material of thickness *l* is given by Beer’s law
(4)$$\frac{dP}{d\omega }\left(l\right)={\left(\frac{dP}{d\omega }\right)}_{0}e^{-\alpha (\omega )l},$$
where *α*(*ω*) is the attenuation coefficient, which depends on the refractive index and the absorption cross section of the material [8].

A further attenuation of the X-ray emission occurs along the propagation in the air between the processed workpiece and a given detector. The influence of the surrounding atmosphere can again be described with Equation (4) by adapting *α*(*ω*) and *l*. Because air at atmospheric pressure is not transparent for soft X-ray photons with energies less than 2 keV, these photons will not reach the detector under normal industrial conditions.

The reduction in the detectable ionizing radiation due to the changing hole geometry is referred to as “self-shielding” of the process in the following, and the corresponding model is discussed in Section 3.3.

## 3. X-ray Emission from Percussion Drilling 

### 3.1. Pricinple of Detection

The experiments were performed using a Ti:Sapphire laser (Spitfire ACE, Newport Spectra Physics GmbH, Darmstadt, Germany) which provides pulses with adjustable duration ranging from 35 fs to 6 ps, an average output power of up to 7 W, and a wavelength of 800 nm (FWHM = 30 nm). The pulse repetition rate was 1 kHz, resulting in pulse energies of up to 7 mJ. The beam had a raw diameter of 10 mm and was focused by means of an F-Theta lens with a focal length of 400 mm to a spot diameter of 2*ω*_0_ = (48 ± 3) µm on the sample’s surface, as shown in Figure 3a. The diameter of the beam waist was experimentally verified by the method of Liu [29]. If not differently stated, a pulse duration of 1 ps was used, resulting in a calculated peak irradiance *J*_0_ of 1.1·10^14^ W/cm^2^. The focal z-position was kept at the original surface of the sample during the processing. All experiments were performed under industrial conditions using a sample made of stainless-steel (1.4301). In order to avoid an influence of polarization, a quarter-wave plate in combination with a half-wave plate was used to obtain circularly polarized radiation for processing. A total of 20 holes were drilled during the experiment. Each data point *i* was obtained by processing all holes with the same number *N_i_* of pulses. The number of pulses was 25 for 1 ≤ *i* ≤ 2, 50 pulses for 3 ≤ *i* ≤ 5, and 100 pulses for 6 ≤ *i* ≤ 14, reaching up to 1000 pulses for each hole. The experimental principle is visualized in Figure 3b, where *N*_1_ and *N*_2_ exemplarily denote the number of pulses applied during measurement *i* = 1 and *i* = 2.

A spectrometer (XRS Detector System, PN Detector, München, Germany), was chosen for the detection of the X-ray photons since the number of emitted X-ray photons was expected to be small due to the short processing time. The spectrometer had an active detection area of 30 µm × 30 µm, with a thickness of 450 µm, which was covered with an 8 µm thick beryllium foil to keep it light-tight. The spectral response of the spectrometer covers the soft X-ray range of interest here and can be found in [30]. The spectrometer was placed at a distance of 35 cm to the processing area in order to keep the pile-up effects [31] low. The detection angle *β* (Figure 2 and Figure 3a) varied between 15° and 70°.

### 3.2. Experimental Results

Figure 4 shows the X-ray yield, which is defined as the sum of the detected X-ray photon energies divided by the total incident laser energy, as a function of the aspect ratio of the drilled hole for three different detection angles β. The aspect ratio $\xi =z_{h}/d_{h}$ is defined by the hole depth *z_h_* and the diameter *d_h_* of the hole at the surface of the samples.

The yield of the X-ray emission shows a clear dependence on the detection angle. At the beginning of the drilling process, where no shielding by the surrounding material occurs, the maximum yield is detected at angles *β* ≥ 35°. This finding is consistent with the results reported by Legall et al. for detection angles of up to 40° [7]. The data for 35° and 70° in Figure 4 show a moderate increase in the X-ray yield with an increasing aspect ratio of the drilled hole before a significant decrease sets in. The initial increase may be explained by the onset of multiple reflections in the deepening hole, resulting in an increased local irradiance and thus an increased X-ray emission. For *β* = 70°, this enhanced X-ray emission results in an increase in the yield by a factor of 2 compared to the yield during processing of a plane surface. With further deepening of the hole, the X-ray yield decreases until a final level is reached, which is independent on the detection angle. From geometrical considerations, it follows that the X-ray yield should decrease due to shielding of the ionizing radiation by the surrounding material of the drilling hole as soon as ξ ≥ 0.5·tan(*β*), which is in excellent agreement with the experimental data. 

The non-vanishing signal detected even with deep holes is assumed to arise from defocusing effects from the plasma on the laser beam, which leads to higher irradiance near the edges of the entrance of the hole [32]. This effect can lead to emission of X-ray photons from a region of the surface next to the hole where no shielding occurs.

The error bars correspond to the standard deviation of the aspect ratios of the 20 holes which were drilled with the same parameters. The diameter of the holes was measured by means of a laser scanning microscope (VK-9700, Keyence Corporation, Osaka, Japan), and the depth of the holes by means of optical coherence tomography (CHRocodile, Precitec, Gaggenau-Bad Rotenfels, Germany). The errors bars of the yield (vertical width of colored areas) are one standard deviation of the distribution of all detected X-ray photon energies.

### 3.3. Calculation of Self-Shielding

An analytical model is presented in the following to calculate the expected X-ray radiation at a detector’s location. The hole is assumed to have a conical geometry, as sketched in Figure 2. The local irradiance inside the hole was calculated by means of raytracing, allowing for determination of the emission of X-radiation individually for each point on the walls inside the hole.

Starting by using Equations (1)–(3) and $V_{P}=dA\cdot \left(l_{a}+c_{S}\tau_{P}\right)$ for the corresponding plasma volume for each infinitesimal surface element *dA* in the hole, where $l_{a}$ is the optical penetration depth, $c_{S}$ is the ion’s speed of sound, and $\tau_{P}$ is the pulse duration of the laser pulse, the power emitted per unit frequency and per unit area can be calculated for each depth *z = z_val_* inside the hole by
(5)$${ℇ}_{0}\left(\omega ,z\right)=\frac{d^{2}P_{B}}{d\omega dA}=\frac{\left(l_{a}+c_{S}(z)\tau_{P}\right)32\pi }{3}\sqrt{\frac{2\pi }{3m_{e}k_{b}T_{h}(z)}}\frac{Z_{i}{\left(z\right)e}^{6}{(n}_{h}{\left(z\right))}^{2}}{m_{e}c^{3}}e^{-\frac{\hbar \omega }{{k_{b}T}_{h}(z)}}.$$ The degree of ionization *Z_i_*, the ion speed of sound $c_{S}$, and the number density of the hot electrons *n_h_* are calculated for each depth by
(6)$$Z_{i}\left(z\right)=Z_{E}\cdot atan\left[{\left(Z_{E}\frac{\frac{3}{2}k_{b}T_{h}\left(z\right)}{3E_{i,max}}\right)}^{\frac{1}{3}}\;\right],$$
(7)$$c_{s}\left(z\right)=\sqrt{\frac{Z_{i}\left(z\right)k_{b}T_{h}(z)}{m_{i}}}$$
and
(8)$$n_{h}\left(z\right)=q_{h}\cdot Z_{i}\left(z\right)\cdot \frac{\rho }{m_{i}}\cdot \frac{l_{a}}{l_{a}+c_{S}(z)\tau_{P}}\;,$$
respectively [18], where *Z_E_* is the atomic number, $E_{i,max}$ is the maximum ionization energy of the element, ρ is the density of the material, and *m_i_* the mass of the ions. The factor *q_h_* denotes the fraction of the free electrons which are “hot electrons”.

The evaluation depth *z_val_* further determines the thickness

(9)$$s_{shield}\left(z_{val},z_{h},\xi ,\beta \right)=\frac{z_{cut}\left(r_{cut}\left(z_{val},z_{h},\xi ,\beta \right),z_{h},\xi \right)}{\sin\left(\beta \right)}.$$
of the surrounding material that the X-ray radiation needs to transradiate to reach a given detector position, where
(10)$$r_{cut}\left(z_{val},z_{h},\xi ,\beta \right)=\frac{{(z}_{val}-z_{h})\cdot \left(1+\frac{\tan\left(\beta \right)}{2\xi }\right)}{2\xi -tan(\beta )},$$
and
(11)$$z_{cut}\left(r_{cut},\xi \right)=2\xi r_{cut}.$$
refer to the position, where the emitted photons from an evaluated emission depth *z_val_* hit the opposite wall of the drilling hole, see Figure 2. The area-specific spectral power of the X-ray radiation transmitted through the material is finally found by inserting (5) and (9) into (4) which leads to
(12)$${ℇ}_{s}\left(\omega ,z_{val}\right)={ℇ}_{0}\left(\omega ,z_{val}\right)\cdot e^{-\alpha \left(\omega \right)\cdot s_{shield}\left(z_{val},z_{h},\xi ,\beta \right)}.$$The energy-specific attenuation coefficients *α*(*ω*) for different materials are provided by the National Institute of Standards and Technology (NIST, Gaithersburg, MD, USA) [33]. The reflection of the X-ray radiation on the walls of the hole was neglected, as this fraction is very small [8].

The total spectral power transmitted through the surrounding material was calculated by integrating ${ℇ}_{s}\left(\omega ,z_{val}\right)$ over the walls of the hole up to the depth *z_h_*, 

(13)$$E_{s}\left(\omega ,\;z_{h}\right)=\pi \frac{d_{h}\sqrt{z_{h}^{2}+{\left(d_{h}/2\right)}^{2}}}{z_{h}^{2}}\int_{0}^{z_{h}}{ℇ}_{s}\left(\omega ,z\right)\cdot (z_{h}-z)dz.$$ Once transmitted through the material, the radiation is further attenuated in the air, hence
(14)$$E_{d}\left(d_{det},\omega ,z_{h}\right)=\vartheta_{air}\left(d_{det},\omega \right)\cdot E_{s}\left(\omega ,z_{h}\right),$$
where $\vartheta_{air}$ is the energy-specific transmissivity through air and the distance $d_{det}$ to the detector may be assumed to be virtually independent of $z_{val}$, as the dimensions of the holes are much smaller than the distance to the detector. The expected average power of the X-ray radiation reaching a detector’s position can now be calculated by integrating Equation (14) over the frequency *ω* and the hole depth *z_h_* > 0,
(15)$$P\left(d_{det},z_{h}\right)=\pi \frac{d_{h}\sqrt{z_{h}^{2}+{\left(d_{h}/2\right)}^{2}}}{z_{h}^{2}}\cdot \int_{\omega =0}^{\omega =\infty }\vartheta_{air}\left(d_{det},\omega \right)\cdot \int_{z=0}^{z=z_{h}}{ℇ}_{s}\left(\omega ,z\right)\cdot (z_{h}-z)dzd\omega .$$
Figure 5a shows the yield, calculated with Equation (15) for the three detection angles which were investigated during the experiments. All parameters used for the calculations are shown in Table 1.

While irradiating very shallow holes with ξ≈0.15 right at the beginning of the drilling process, the model predicts nearly the same yield for all detection angles. For small observations angles, *β* < 35°, the yield immediately starts to diminish right from the beginning of the drilling process due to the shielding effect of the material surrounding the hole. At higher detection angles, here seen with the calculations for *β* = 70°, the detected X-ray emission is first found to increase before it drops again when the hole reaches a certain aspect ratio, as expected due to the shielding by the surrounding material. The initial enhancement arises due to the increase in local irradiance *J*(*z*) inside the drilling hole, which results from multiple reflections of the laser pulse incident in the hole. This effect leads to an increase in efficiency of the X-ray emission by a factor of more than 3 for a detection angle of 70° or above.

The experimental results, mentioned in the previous section, showed a small remaining signal for higher aspect ratios, which was nearly identical for all three investigated detection angles. This signal is assumed to arise from the influence of a plasma, which is formed near the entrance of the drilling hole, on the laser beam. The plasma can lead to defocusing of the laser beam [32], and thus lead to an increased irradiance near the edges of the entrance of the hole. X-ray emission from these regions can reach the detector without further shielding by the bulk material around the hole. This additional X-ray emission of the unshielded plasma near the surface of the sample was considered by adding a fraction $c_{u}$ of the expected emission that would be generated when processing a plane surface with the laser parameters used during the experiments. The local irradiance was again obtained from the raytracing calculations. This unshielded part of the X-ray emission is independent of the detection angle and was assumed to be constant during the whole drilling process. The value of $c_{u}=0.05$ was found using a best fit method. The power $P_{u}$ of the unshielded emission was calculated by replacing z with the radial coordinate r in the expression for the X-ray emission given by Equation (5) and integrating it over a circular area which corresponds to the opening of the hole on the surface of the samples, finally integrating over the spectrum while considering the attenuation of the radiation in the air
(16)$$P_{u}\left(d_{det}\right)=c_{u}\int_{\omega =0}^{\omega =\infty }\vartheta_{air}\left(d_{det},\omega \right)\cdot \int_{r=0}^{r=r_{h}}{ℇ}_{0}\left(\omega ,r\right)2\pi r\;drd\omega .$$ It is worth noting that the same result is obtained by substituting the integration variable dz in (15) with dr using $r\left(z\right)=d_{h}/2-z\cdot d_{h}/2z_{h}$ and setting $z_{h}=0$ in the result (hole with zero depth corresponds to flat surface).

The results shown in Figure 5b are the sum of Equations (15) and (16) for the three evaluated detection angles (as well as the unshielded emission itself) following Equation (16), depicted in green.

## 4. Discussion

The results of the measurements are compared to the corresponding model calculation in Figure 6 separately for each of the three considered detection angles. It can be stated that the predictions of the model are in good agreement with the measured data for all validated detection angles. At *β* = 15° (Figure 6a) the shielding effect of the surrounding material of the hole is slightly weaker than predicted by the model, but still within the experimental uncertainties discussed in Section 3.2. For detection angles elevated higher above the surface, the increase in yield and, subsequently, the aspect ratio at which the maximum occurs are predicted very well by the model. The remaining yield at high aspect ratios is well-described by an unshielded plasma emission originating from the upper part of the hole and can reach the detector without any further shielding of the surrounding solid material. Finally, both, experimental data and model calculations showed a decrease in the X-ray yield by at least one order of magnitude with an increasing aspect ratio until ξ=3.

For the experiments, the applied peak irradiance was calculated to amount to 1.1·10^14^ W/cm^2^. This irradiance is two–three orders of magnitude higher than the ablation threshold for the used pulse duration of 1 ps and thus at least one order of magnitude above the irradiances, which are typically used to drill holes with good quality in practice. Using industrial laser systems, which operate at a repetition rate of several tenths or hundredths of kHz, the evaluated aspect ratios are reached within processing times of less than one second. One can conclude from the findings reported above that X-ray emission during percussion drilling is only relevantly detectable at the very beginning of each drilled hole. 

## 5. Summary

A model to calculate X-ray emission from laser percussion drilling with ultrashort pulses was presented and evaluated with data from corresponding experiments. The experiments were performed using a Ti:Sa laser with a wavelength of 800 nm, a pulse duration of 1 ps, a pulse frequency of 1 kHz, and a pulse energy of 1 mJ. Percussion drilling with a total of up to 1000 pulses for each hole was performed on a sample of stainless steel (1.4301) with a peak irradiance of 1.1·10^14^ W/cm^2^ on the plane surface. The X-ray emission was measured with a spectrometer for three different detection angles at a distance of 35 cm from the processing zone. The model calculations of shielding effects due to the solid material surrounding the hole are in very good agreement with the experimental data. 

## 6. Conclusions

The model is suitable for predicting the self-shielding of X-ray emission from ultrafast laser processing due to geometrical changes of the interaction zone.

By combining the given model with other analytical models, the presented method can be adapted to other irradiation conditions. This includes, e.g., the progress of laser drilling in conical holes, as described in [27]. Furthermore, the model allows for describing the self-shielding when processing other metals by adapting the corresponding transmission coefficients [8] for the X-ray transmission through the material. However, the presented formulas are only valid for metals. Calculating the self-shielding during processing of semiconductors and dielectrics would require modification of Equations (3) and (5) while considering non-linear absorption.

The presented work can be used to estimate the reduced potential hazard during processing of deep structures. One major finding, regarding safety issues due to ionizing radiation, was that X-ray emission during percussion drilling is only relevantly detectable at the beginning of the drilled hole until there are no shielding effects due to the surrounding material. The curse of the dose rate, which would result from the investigated drilling process, follows the same curse as the yield shown in Figure 6. These normalized dose rates were measured to be below 1 (µSv/h)/W due to the short processing time and number of pulses which it takes to reach aspect rations ξ>3. At these points, the yield has already decreased by at least one order of magnitude compared to its initial value.

Future investigations might include space resolved measurements of the locations of the X-ray emission. This would allow for explaining the reason of the remaining yield at high aspect rations.

## Figures and Tables

**Figure 1 materials-17-01109-f001:**
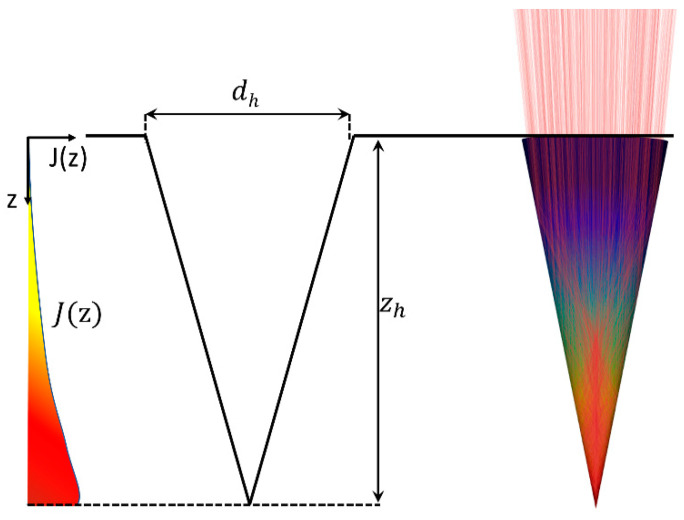
Model geometry of the percussion-drilled hole with a qualitative sketch of the distribution of the irradiance J(z). As a consequence of multiple reflections inside of the hole, as indicated by the red rays (**right**), the distribution of the irradiance J(z) exhibits a maximum near the tip of the hole (**left**).

**Figure 2 materials-17-01109-f002:**
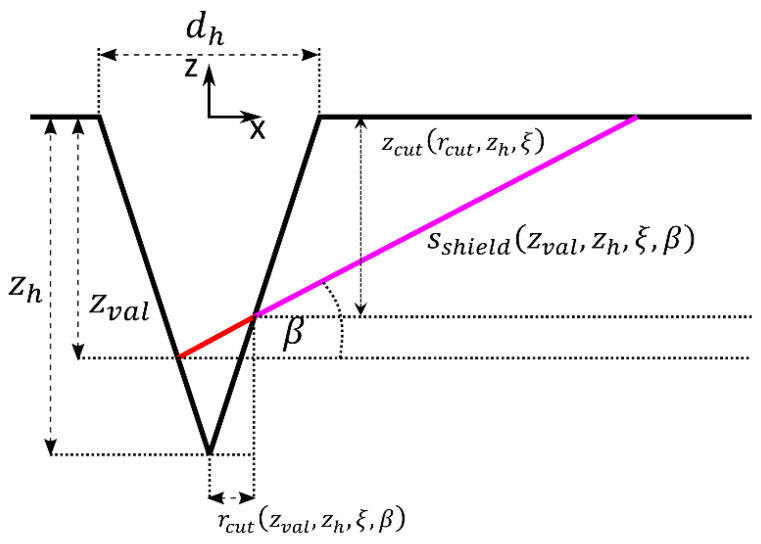
Geometrical parameters used to model the spectral power of the X-ray emission detectable at a given point outside of a percussion-drilled conical hole.

**Figure 3 materials-17-01109-f003:**
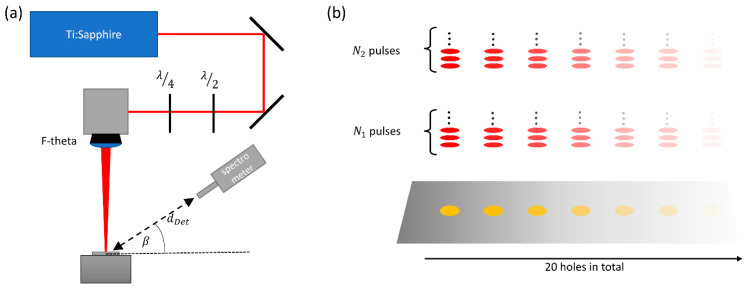
Experimental setup. (**a**) The beam of a Ti:Sapphire laser with a wavelength of 800 nm and a pulse repetition rate of 1 kHz was focused on a sample of stainless steel by means of an F-theta lens, leading to a peak irradiance of 1.1·10^14^ W/cm^2^ on the initially plane surface. The spectrometer was placed at a distance of 35 cm from the processing region. (**b**) A total of 20 holes were processed for each measurement. All holes were repeatedly processed with the same number *N_i_* of pulses, and the X-ray emission was measured separately for each processing step *i*. Hence, every measurement corresponds to the emission from a specific depth *z_h_* of the drilled holes.

**Figure 4 materials-17-01109-f004:**
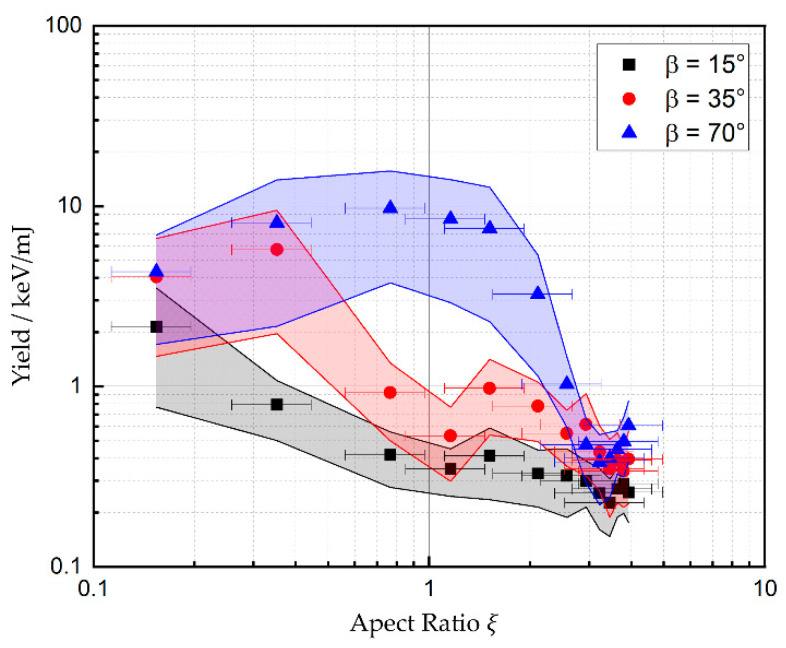
Yield of the detected X-ray emission from percussion-drilled holes with increasing aspect ratio. The detector was placed at a distance of 35 cm from the processing zone. Each data point corresponds to the average over the 20 holes drilled with the same parameters.

**Figure 5 materials-17-01109-f005:**
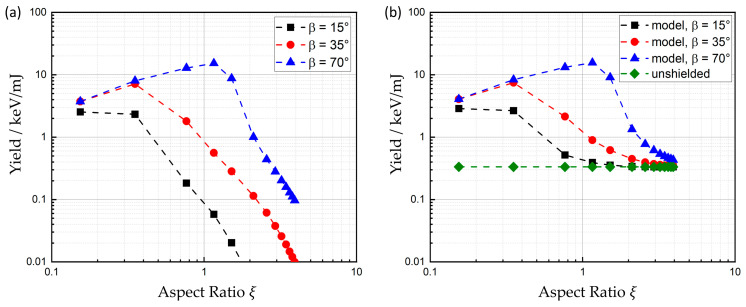
Calculated X-ray emission reaching a detector placed in 35 cm from the drilled holes in the direction β. (**a**) Results according to Equation (14). (**b**) Results from Equation (14) with further details, including an unshielded emission from the surface next to the hole (green dotted line) following Equation (15).

**Figure 6 materials-17-01109-f006:**
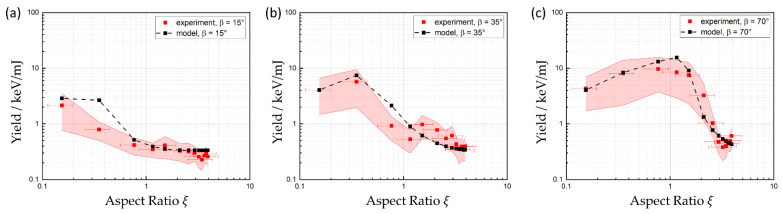
Comparison of measurements and model calculation. Depending on the detection angle, the yield of X-ray emission first increases (**b**,**c**) before dropping to a constant value, which is identical for all three detection angles. For small detection angles (**a**) there is no increase at the beginning visible, because the discussed shielding effect starts immediately.

**Table 1 materials-17-01109-t001:** Values of the parameters, which were used to calculate the X-ray yield.

Parameter	Symbol	Value
Laser beam profile		Gaussian
Pulse duration	*τ_p_*	1 ps
Beam power	*P*	1 W
Laser pulse repetition rate	*f_rep_*	1 kHz
Beam waist radius (1/e^2^)	*ω* _0_	24 µm
Beam raw diameter	*d* _0_	10 mm
Focal width (F-Theta lens)	*f_L_*	400 mm
Laser wavelength	*λ*	800 nm
Atomic number (Material)	*Z_e_*	26 (Iron)
Mass of ions	*m_i_*	9.274·10^−26^ kg [34]
Complex refractive index	*n*, *k*	*n* = 2.8, *k* = 3.36 [35]
Material density	*ρ*	7.874 g/cm^3^ [36]
Hole diameter	$d_{h}$	90 µm
Distance of detector	*d_det_*	35 cm
Angle of detection	*β*	15°, 35°, 70°
Optical penetration depth	*ℓ_a_*	19 nm [35]
Max. ionization energy	*E_i,max_*	9278 eV [37]
Fraction of hot electrons	*q_h_*	5.5·10^−4^ [18]

## Data Availability

Data are contained within the article.

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
