# Peer review of "Self-Shielding of X-ray Emission from Ultrafast Laser Processing Due to Geometrical Changes of the Interaction Zone"

_materials, 2024, doi:10.3390/ma17051109_

Round 1

Reviewer 1 Report

Comments and Suggestions for Authors

This paper presents a study of soft x-ray emission under standard irradiation conditions, applicable to both industry and a research laboratory. A model has been proposed for determining the intensity and geometry of the X-rays emitted during the irradiation.

This work contributes with a very specific knowledge, that can be relevant in the field, but with limited application range. The topic of X-ray generation has been extensively studied, with very detailed works. The most relevant are included in the references. Respect to these publications, this work contributes with a theoretical model that can predict the emission in a very specific experimental case.

The introduction clearly sets the context in which the work is framed and outlines the main objectives being addressed. The paper follows a clear and precise structure, making it highly suitable for a good understanding of the work. In addition, the mathematical development is appropriate and explained in an accurate way, enabling reader to follow and reproduce it if necessary easily. The novelty of this work is based mainly in the simulation method proposed. To prove the accuracy of the method, the authors develops a series of experiments that match with the simulation.

However, even that the mail goal of the work is clearly explained, I miss a brief explanation of how to adapt the method to other irradiation conditions. The paper shows the results in stainless steel in a static position, being a very limited process condition respect to the techniques used in industry/research. What would happen in other materials? Dielectrics, for example, will require higher doses of energy creating different hole shapes (and probably can produce less amount of X-rays). These casuistries are covered by the method? Can be adapted easily or new studies have to be done? The uses of new materials, or even, dynamic irradiation, could modify significantly the geometry of the “cone”. I suppose authors have chosen this condition and material to favoring the creation of X-ray, but in order to extrapolate the method to other application some words regarding this can be added.

The description of the experimental method followed for the irradiation seems a little bit confusing (lines 160-165). I don’t clearly understand the number of pulses in each step, did you make 20 holes with different pulse numbers steps until achieving 1000 pulses? Other important parameter in my opinion in the context of this work is the pulse duration. What is the expected influence of shorter pulse duration? The geometry of the cone could be modified with this parameter.

As mentioned previously, the text is easy to follow and is well explained. The figures and captions are correct and give the precise information to understand each figure. The discussion and the conclusions are in good agreement with the results. In my opinion, this a relevant paper that could be applied directly conducting risk analysis in both industry and research institution.

Reviewer 2 Report

Comments and Suggestions for Authors

Dear Authors

The presented article addresses the important issue of occupational safety, especially the emission of X-rays during laser processing. Because the use of lasers is increasing not only in industry but also in other areas, the issue considered in the article is also of great importance in the future. I evaluate this work positively, but I post a few minor comments below.

1. In chapter 4 (Line 284, 285) it is written "...but still within the experimental uncertainties." At this point in the article, please provide information about what "experimental uncertainties" are, how much they are, and what the numerical criteria are.

2. The last sentence in chapter 4 (Lines 298-300) is an important conclusion that should be included in chapter 5 because sometimes readers, due to lack of time, do not analyze the entire article but read mainly the summary and conclusions.

3. Conclusions (chapter 5) are more like a summary of the article. I propose to redraft the text in Chapter 5 to better highlight the importance of the research conducted for work safety in industrial and laboratory conditions.

Kind regards

Reviewer

Reviewer 3 Report

Comments and Suggestions for Authors

This work develops self-shielding as a way to limit or eliminate hazardous X-ray emission occurred in the course of ultrafast laser processing of steel, due to geometrical changes of the interaction zone. The research is well-designed and presented clearly. A good comparative analysis of existing publications and the tasks set in the work is carried out. The methodological section of the manuscript is presented in sufficient detail. The authors used modern equipment for the preparation and testing of samples. They also utilized the equipment for visualization and assistance in interpreting the obtained results. They calculated the expected X-ray radiation at a detector location. The results of the experimental measurements were compared to the corresponding model calculation. Large statistical samples confirm the reliability of the presented experimental data. The authors showed that the predictions of the model are in good agreement with the measured data for all validated detection angles (15°, 35°, and 70°). The model is suitable to predict the magnitude of potentially hazardous ionizing radiation from laser percussion drilling and laser cutting processes using ultrashort pulses.

However, some shortcomings should be corrected to make the manuscript acceptable for publication in Materials.

(1) Lines 37–38: The phrase “electron temperatures in the order of keV” should be corrected scientifically in terms of either “electron temperatures” or “keV”.

(2) It is inappropriate to use the same “T” symbol for different parameters (Lines 100-105 vs. Lines 234-240).

(3) Figure 5 and Figure 6 are numbered wrongly. This should be fixed in the captions and the main text.

(4) The style of presentation of Figures is somehow unusual. It is common practice to use “Figure 6(a)”, “Figure 5(b,c)” instead of “Figure 6, left”, “Figure 5, middle and right”.

(5) Figure 6 should be placed after the first mention of it, i.e. in the Discussion section.

Reviewer 4 Report

Comments and Suggestions for Authors

1) The paper solidly presents the simulation model and experimental method for verification. The tables and graphs are intuitive and instructive.

2) What is missing is the conclusive relation to existing regulatory norms for X-ray radiation exposure limits for industrial workers. There is no calculated value for the radiation dose and dose rate. The expression “relevantly detectable” leaves the reader in the dark about the question if the radiation level is relevant for occupational health and safety (OHS) limits. 

3) Since only the onset of hole drilling is “relevantly detectable” in the X-ray detector, there is an element of dynamics which should enter quantitatively into the assessment of radiation exposure relevance (--> dose rates).

4) Another essential information is missing: the details about the spectrometer, it's spectral sensitivity and the actually detected spectral distribution (at how many keV was the peak and range of the emission?). This info would also enter any useful dose calculations.

5) Abstract: Is there some main quantitative (numerical) result of this study? Adding this into the abstract would strengthen the manuscript.

6) Finally two minor editorial / language comments:

Line 41: The transmission even of 5 keV x-rays through air at these ambient conditions at half meter distance is only ~ 9%. That’s intolerable for an operator, but certainly not “transparent”. Wording should be improved (or clearly define the use of the term “transparent”, see also line 51).

Line 199-201: The sentence is grammatically wrong and misses a dot at the end.
